# The Effectiveness of Online Platforms after the Pandemic: Will Face-to-Face Classes Affect Students' Perception of Their Behavioural Intention (BIU) to Use Online Platforms?

Rana Saeed Al-Maroof [1], Noha Alnazzawi [2], Iman A. Akour [3], Kevin Ayoubi [4], Khadija Alhumaid [5], Nafla Mahdi AlAhbabi [6], Maryam Alnnaimi [1], Sarah Thabit [7], Raghad Alfaisal [8], Ahmad Aburayya [9] and Said Salloum [10,*]

1　English Language & Linguistics Department, Al Buraimi University College, Al Buraimi 512, Oman; rana@buc.edu.om (R.S.A.-M.); maryam@buc.edu.om (M.A.)
2　Department of Computer Science and Engineering, Yanbu University College, Yanbu 30436, Saudi Arabia; alnazzawin@rcyci.edu.sa
3　Information Systems Department, University of Sharjah, Sharjah 26666, United Arab Emirates; aalhamad@sharjah.ac.ae
4　Department of English/General Studies, Khalifa Bin Zayed Air College, Al Ain 295, United Arab Emirates; kayoubi@hct.ac.ae
5　College of Education, Zayed University, Abu Dhabi 19282, United Arab Emirates; khadija.alhumaid@zu.ac.ae
6　Training and Professional Department, Emirates Schools Establishment (ESE), Abu Dhabi 971, United Arab Emirates; nafla.alahbabi@ese.gov.ae
7　Linguistic Department, School of Humanities, Universiti Sains Malaysia, Gelugor 11800, Malaysia; sarahthabit@student.usm.my
8　Computing and Creative Industries, Faculty of Art, Universiti Pendidikan Sultan Idris, Tanjong Malim 35900, Malaysia; raghad.alfaisal81@gmail.com
9　Quality & Corporate Development Office, Dubai Health Authority, Dubai 9115, United Arab Emirates; amaburayya@dha.gov.ae
10　School of Science, Engineering, and Environment, University of Salford, Salford M5 4WT, UK
*　Correspondence: salloum78@live.com

**Abstract:** The purpose of this study is to investigate students' intention to continue using online learning platforms during face-to-face traditional classes in a way that is parallel to their usage during online virtual classes (during the pandemic). This investigation of students' intention is based on a conceptual model that uses newly used external factors in addition to the technology acceptance model (TAM) contrasts; hence, it takes into consideration users' satisfaction, the external factor of information richness (IR) and the quality of the educational system and information disseminated. The participants were 768 university students who have experienced the teaching environments of both traditional face-to-face classes and online classes during the pandemic. A structural equation modelling (SEM) test was conducted to analyse the independent variables, including the users' situation awareness (SA), perceived ease of use, perceived usefulness, satisfaction, IR, education system quality and information quality. An online questionnaire was used to explore students' perceptions of their intention to use online platforms accessibly in a face-to-face learning environment. The results showed that (a) students prefer online platforms that have a higher level of content richness, to be able to implement the three dimensions of users' situation awareness (perception, comprehension and projection); (b) there were significant effects of TAM constructs on students' satisfaction and acceptance; (c) students are in favour of using a learning platform that is characterised by a high level of educational system quality and information quality and (d) students with a higher level of satisfaction have a more positive attitude in their willingness to use the online learning system.

**Keywords:** information richness; information systems; situation awareness; technology acceptance model; online platform

## 1. Introduction

In recent years, there has been a tremendous focus on increasing the acceptance of online learning platforms, as these are used as media to create a new educational environment that has features, tools [1–3] and an atmosphere that is rather different from that of traditional classrooms [4,5]. This investigation aimed at exploring the impact of certain factors on teachers' and students' perceptions of these online learning platforms [6–9]. During to the pandemic, many countries have faced difficulties in shifting to a digitalised world. This shift would lead to a total change in the educational setting, such as that the students and teachers are in different locations but can still interact with one another [10,11]. The situation started to change again after vaccines were made available, and now, most colleges and universities are returning to the traditional classroom. Face-to-face communication will take prevalence once more. This invites a question on the effectiveness of online learning platforms in the future.

Although many studies have tackled the importance of e-learning platforms in educational environments, exploring students' perception [6,12–17], few have investigated the effectiveness of online classes once traditional classes are resumed, with emphasis on the information richness (IR) of the three dimensions of situation awareness (SA). SA can be defined as the perception of information that ensures that users can interpret the content presented in an online class. Then, the comprehension deals with the integration of the information previously absorbed with new information and, finally, the comprehension of the total information made available to achieve the goals of these online classes [18]. Based on the previous assumption, this study aims to examine student and teachers' acceptance of e-learning platforms by:

1. Assessing the quality and the development of e-learning platforms after the pandemic, as they have an impact on students' performance, innovation and SA.
2. Measuring the increased need for e-learning platforms due to their ease of use and usefulness and their impact on academic services, pedagogies and practice for lifelong learning.

SA embraces another dimension that goes beyond perception in order to help users combine, interpret and store information. The perception of information can act as a helpful first step, but the need for the comprehension of the information is crucial. The comprehension has to do with the integration of pieces of information based on the participants' own goals. Projection can be defined as the ability to predict future events that enhance the capability to better understand the new situation [18]. In other words, SA is related to users' ability to take action based on the perception of information that can predict certain future events. Accordingly, SA is related to how environmental elements are perceived, comprehended and predicted [19,20]. Perception is the first level in SA, which has to do with the observation of information. This includes examining the dynamics, features and status that are relevant to the situation. The second level of SA is comprehension, where information is processed. The information is integrated to complete the picture from various perspectives. Comprehension leads to a better understanding of the environment. Finally, projection is the third level, where the reaction is built on time. The project helps predict future events and actions in relation to specific situation comprehension [21].

Accordingly, the conceptual model investigates the effectiveness of e-learning platforms from two different perspectives. The first perspective is related to mere acceptance of technology, where the technology acceptance model (TAM) is used to determine the acceptance of the technological aspects of e-learning platforms by examining their usefulness and ease of use. On the other hand, the second perspective focuses on SA, which investigates the effect of the teaching–learning environment on users' level of acceptance.

In short, studies on online learning platforms have relied mostly on the TAM [22], flow theory (FB) and the unified theory on acceptance and use of technology model (UTAUT) to explore the effectiveness of e-learning platforms before and after the pandemic [23,24]. No known study has explored through a conceptual model the intention to use online learning platforms when face-to-face classes are resumed. The model takes into account

the factor of IR with SA. The theory of SA includes three dimensions, namely perception, comprehension and projection. This study is motivated by the need to fill this knowledge gap. Previous works of research show that TAM constructs and the external factors of enjoyment, familiarity, innovation and personality are all crucially significant in shaping learners' attitudes and perception towards the acceptance of online learning platforms [25–29]. This study is driven by the fact that there is a need to validate the intention to use online learning platforms simultaneously with face-to-face classes in the Gulf area, depending on the conceptual model that comprises TAM constructs and its relation with IR and satisfaction.

## 2. Literature Review

The literature review is divided into two main parts. Each part represents a period where studies were conducted under two completely different sets of circumstances. The first group includes research papers that focus on the effectiveness of e-learning platforms before the pandemic, taking into consideration the fact that face-to-face classes can help as a tool in everyday education. E-learning platforms were used as a secondary tool, and not an essential one, in transmitting information among teachers and learners [25,30,31]. The second group of research papers, however, adopted different strategies that stem from the fact that the pandemic imposed certain new restrictions that made face-to-face interaction impossible. One of the striking differences between the two groups is that the studies conducted before the pandemic examined the acceptance of technology before its actual use. Gradually, studies began focusing on the acceptance of e-learning platforms, as they were used in the educational environment during the pandemic [7,32,33]. The acceptance and adoption of online learning platforms has been the concern of many researchers starting from 2006. The implementation of different models reflects the urgent need to tackle the impact of this technology on students' and teachers' perceptions and their actual use. The variations of the adopted model reflect the fact that these technologies can be investigated by tackling variant models, including TAM, FB and UTAUT [25,30,34]. An investigation of the effectiveness of any learning platform before the pandemic focused on important external factors such as gender, cultural differences, personality differences, technical support, technology training, equipment accessibility and experience [31,34,35]. Furthermore, the importance of flow theory and perceived enjoyment was also evident from these studies. The main reason for this is that these factors can boost the chances of continuing to use these platforms post pandemic. They can work collaboratively with the other models to give deeper insights and more detailed results on the impact of these factors on the teaching–learning process [27,30,34].

The difficult pandemic days compelled teachers and students to shift from the traditional physical classroom, where face-to-face communication is the most common way of interacting, to a kind of digitalised world that depends on different websites and applications. Studies conducted after the outbreak of the pandemic have tackled the issue of acceptance of online learning platforms depending on different models including TAM, TBP and UTAUT, along with other external factors. These studies have shown that most TAM constructs, especially the perceived ease of use and the perceived usefulness, have a high impact on the intention to use technology during the pandemic [13,32]. On the other hand, other studies have ignored the importance of the TAM constructs by examining performance expectancy (PE), effort expectancy (EE), social influence (SI) and social isolation and their effect on online learning platforms. Studies have proven that there is a huge effect of system quality and information quality on students' perception of the actual use of online platforms [26].

The external factors that are considered crucial and effective for the acceptance of online learning platforms tend to vary. The most important factor is the fear of COVID-19 during the pandemic. In addition, there are other external factors such as perceived risk, satisfaction, attitude and innovation [13,33,36]. Surprisingly, the external factors may vary in their level of impact. In a study by [7], the results illustrated that the best predictor for student motivation is enjoyment, which is followed by self-efficacy. In another study

by [37], less emphasis has been placed on the perceived ease of use in comparison with other TAM constructs.

Finally, it is worth mentioning that these results were obtained from university students with different majors, including tourism and hospitality [6] and accounting [38]. The focus on students as part of the sample owes to the fact that they are attachable to the technology and are the most frequent users of these online platforms. These studies have been conducted in different places worldwide. Some of these studies have shown interesting results by comparing students from different cultural backgrounds, such as Mexico, Peru, Turkey and the USA [33].

In Table 1, through the repeated use of TAM and UTAUT models, it has been proven that these models are influential in investigating users' acceptance or adoption of various technologies, including e-learning platforms in education. The implementation of these theories in business, health, e-commerce and agriculture has emphasised the importance of these models not only in education but also in other fields [39–42]. The emphasis on the adoption of these models is continually growing. In a recent study by [41], the researchers build a model based on TAM 1, TAM2 and TAM3, with the aim of examining the factors that influence Malaysian small and medium enterprises (SMEs) to adopt mobile commerce.

**Table 1.** Studies before and after the outbreak of the pandemic.

| | Authors and Dates | Online Learning Category | Model | Type of Study | Results Verification |
|---|---|---|---|---|---|
| Studies Before the Pandemic | [30] | Online Learning System | TAM and FB | Adoption | TAM and TPB can predict e-learning adoption positively. |
| | [25] | E-learning Tool | TAM and the Innovation Diffusion Theory | Adoption | TAM, system quality and computer self-efficacy can positively affect students' behaviour |
| | [34] | Online Learning System | Perceived Self-efficacy Perceived Usefulness Satisfaction | Acceptance | Perceived self-efficacy and perceived usefulness have varied effects due to cultural differences and years of experience |
| | [31] | TAM with a Self-regulation Concept. | Online Learning System | Acceptance | TAM constructs can positively affect the acceptance of technology; they are affected by other factors such as personality differences, technical support, technology training and equipment accessibility |
| | [43] | TAM and Satisfaction | Online Learning | Acceptance | TAM constructs may have varied effects on users' satisfaction in their acceptance of online learning systems; gender and diversity have an impact on TAM constructs. |
| | [27] | UTAUT with a Group of External | Online Learning System | Acceptance | UTAUT and factors of self-regulation, computing device ownership and level of familiarity with education-related technologies can positively affect the acceptance of an online learning system |
| | [44] | TAM and External Factors | Online Learning System | Acceptance | TAM, computer self-efficacy, convenience, instructors' characteristics, instructional design and technological factors positively affect the acceptance of technology |
| | [28] | UTAUT | Two Online Learning Environments | Acceptance | The main constructs affect users' acceptance differently; the model has to be revisited |

**Table 1.** *Cont.*

| Authors and Dates | Online Learning Category | Model | Type of Study | Results Verification |
|---|---|---|---|---|
| [35] | TAM with Experience and User-friendliness | Online Course Delivery | Acceptance | TAM, user friendliness and experience have a positive effect on the acceptance of the online learning system |
| [45] | WebCT Online Learning System (OLS) | TAM | Acceptance | The constructs of PRATAM positively affect the intention to use the OLS. |
| [12] | Online Learning Service | TAM | Adoption | Perceived usefulness and perceived ease of use have a positive impact on the adoption of the online learning service |
| [23] | E-learning Environment | TAM and Flow | Acceptance | The flow-on perceived ease of use and perceived usefulness have a positive effect on the actual usage of the e-learning environment |
| [46] | Online Courses | Learning Environmental Expectancy and Self-Regulation in Terms of Metacognition and Motivation | Acceptance | Self-regulation in terms of metacognition and motivation can positively affect online courses, while metacognition and social negatively affect behavioural intention |
| [47] | A Comparison of Two Learning Platforms | TAM | Acceptance | TAM constructs positively affect the acceptance of online learning in Nigeria and the Philippines |
| [48] | Google Classroom | TAM | Acceptance | TAM constructs positively affect the acceptance of online learning in Oman |
| [49] | Google Classroom | TAM and External Factors | Continuous Intention to Use Technology | TAM contracts, along with system quality, information quality and concentration, affect students' satisfaction and intention to use the technology in the future |
| Studies After the Outbreak of the Pandemic [32] | Microsoft Teams | TAM | Acceptance | Perceived usability is highly positive due to the lack of physical classes |
| [6] | Online Classes | TAM and TPB with Innovation as a Moderator | Acceptance | TAM constructs positively affect the acceptance of technology, except PEOU and perceived behavioural control; innovativeness has a moderating role between subjective norms and behavioural intention |
| [33] | Educational System at Universities | TAM and Attitude | Acceptance | The technology is positively evaluated by students' attitude, which is affected by obstacles due to the limited internet environment |
| [7] | Learning Management System | UTAUT | Acceptance | The acceptance of an online learning system is positively affected by PE, EE and SI; COVID-19-related fears help moderate the link of PE and SI with BI |
| [37] | Zoom application | TAM | Acceptance | TAM constructs have a positive effect on the actual use of the online learning system |
| [13] | Online and Mobile Technology | UTAUT with Trust and the perceived Risk | Acceptance | The adoption of technology during the quarantine is positively affected by PE, EE, trust and perceived risk |
| [50] | Google Meet | TAM and the External Factor of Fear of COVID19 | Acceptance | Fear-related factors negatively affect the intention to use, but other TAM constructs help enhance the process of learning |

In short, studies conducted before the pandemic focused on certain external factors that may be different from those emphasised by researchers after the outbreak of the pandemic. However, the implementation of TAM and UTAUT as concrete models is evident, assuming that they can measure the acceptance or adoption of online learning platforms effectively and practically. The table below illustrates the main studies conducted before and after the outbreak of the pandemic with respect to the acceptance and adoption of online learning platforms.

## 3. Methodology and Research Model

A research model, as shown in Figure 1, was developed to examine users' intention to use online platforms with seven hypothesised relationships. This research model includes three important dimensions of SA towards IR, which enabled us to explore the effectiveness of these dimensions on the newly formed conceptual model. The level of IR may influence the online learning platform differently. The TAM constructs, along with the educational quality system and information system, add new values in assessing the effectiveness of the online learning platform in relation to IR. Finally, the factor of users' satisfaction can contribute more effectively to the conceptual model and improve users' intention to use the online platform.

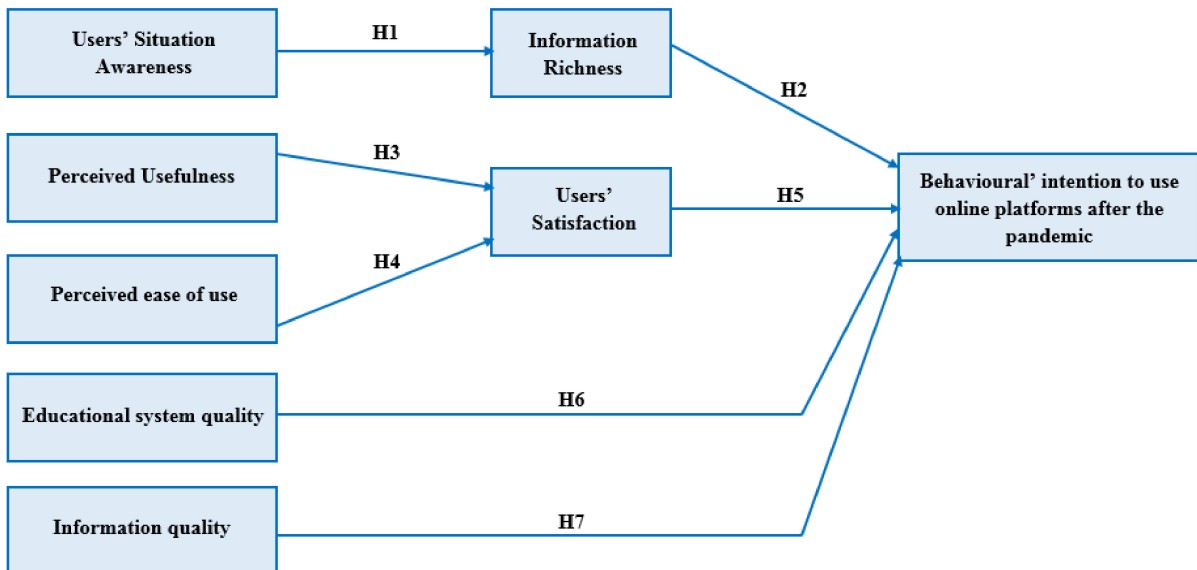

**Figure 1.** Research model.

### 3.1. Situation Awareness and Information Richness

SA addresses the understanding of what is going on and what might happen next. It has been defined as the ability to develop an accurate internal representation of events in the environment that may lead to successful decision-making. The failure to recognise the importance of SA in a specific environment may lead to conceptual confusion and a degree of conceptual tension [51–53]. Similarly, IR is affected by the environment in the sense that it has a strong impact on individuals' behaviour. When the provided information is sufficient, it can act as a tool to enhance an individual's experience, allowing the development of trust. Accurate IR may lead to the creation of a better teaching–learning environment and increase trust in the technology [54,55].

Information content richness has a close relationship with the quality of the received information with respect to certain aspects, including clarity, relevance, sufficiency, accuracy, timeliness and simplicity. There are various forms that can cover users' choices [55,56]. IR is considered a dominant, effective and influential aspect when it comes to online learning platforms, and it has a close relationship with users' personalities, trust and [24,54].

IR is related to SA, which is a key factor to understanding how to decide in new situations. SA is governed by the three dimensions of perception, comprehension and projection. Perception is the first important step in formulating a perfect picture of a situation. It is a fundamental aspect, without which it would be impossible to get a clear picture of the importance of the perceived information. Projection can be defined as the ability to predict future events, which enhances the capability to get a better understanding of a new situation. The experience that is accumulated in the process can deeply enhance the forecast of future events [57].

SA embraces another dimension that goes beyond perception to help users combine, interpret and store information. The perception of information can act as a first step, but the need for comprehending the information is crucial. This comprehension has to do with the integration of the pieces of information based on the participants' own goals. Projection can be defined as the ability to predict future events, which enhances the capability to better understand a new situation [18].

This study has created a connection between the effectiveness of IR and the three dimensions of SA, which are perception, comprehension and projection. Whenever the richness in information is evaluated by users to be clear, comprehendible and easily perceptible, the online platform is considered an effective and highly preferred tool. According to the previous assumptions, the following hypotheses are proposed:

**Hypothesis 1 (H1).** *SA positively affects the IR in using online platforms after the pandemic.*

**Hypothesis 2 (H2).** *The IR positively affects the use of online platforms after the pandemic.*

*3.2. User Satisfaction and TAM Constructs*

User satisfaction is closely related to the users' attitude towards the intention to use technology. According to Dunbar et al. [58], users' satisfaction refers to the impact that an application of technology has on users' attitudes when they are using the technology. Furthermore, Dalvi-Esfahani et al. [59] defined user satisfaction as the type of feeling or pleasure which appears as an outcome of using technology due to its benefits. A similar definition is presented by Dalvi-Esfahani et al. [59], who defines user satisfaction as a subjective evaluation that can be pleasant or unpleasant, proposing that the evaluation appears due to the use of certain technology. Satisfaction has a close relationship with individual attitude, organisation purposes and social consequences. Users' satisfaction is useful in evaluating the effects of online platforms.

On the other hand, perceived ease of use is defined as the degree of effort that users may reflect in using a piece of technology. Whenever the technology is evaluated as effortless, it implies a preference for it, and the intention to use it is evident. Similarly, perceived usefulness is defined as the degree of usefulness that a person derives from a specific system that has a direct effect on his or her performance [60]. Previously, Kaufhold et al. [61] stated clearly that usefulness is the most powerful predictive variable in evaluating information technology usage. The TAM has been used extensively by many studies where usefulness is the key factor in measuring users' intention to use new technology. Both the perceived ease of use and the perceived usefulness are considered influential elements in deterring the effectiveness of technology, and they have a direct effect on users' level of satisfaction. These two constructs are investigated by prior studies to reflect different purposes. For instance, Halimeh et al., and (Krejcie & Morgan) [62,63] consider TAM constructs as variables that are used to identify the relation between e-wallet and mobile banking with customers' perception. Thus, the following hypotheses are formulated:

**Hypothesis 3 (H3).** *The intention to use an online platform is positively affected by the users' perception of ease of use.*

**Hypothesis 4 (H4).** *The intention to use an online platform is positively affected by the users' perception of usefulness.*

**Hypothesis 5 (H5).** *The intention to use an online platform is positively affected by users' satisfaction.*

*3.3. Educational System Quality and Information Quality*

The educational system entails the use of a particular system due to its easiness, content and enjoyable features. This implies that whenever users evaluate a system, such as a difficulty, it entails that the system will not be functionally acceptable. Online learning platforms are affected by the quality of the education system since it enhances the understanding of the course in an educational atmosphere. The concept of the educational system has been expanded to include ways of developing educational profiles, taking into consideration the benefits and effects of developing and implementing new features. The education quality embraces not only the quality of the system itself functionally, but the quality of the information academically [64,65].

Furthermore, information quality is defined as the quality of the content of the information system, including factors such as the intelligibility, objectivity, sufficiency and relevance of the content. This factor can effectively and exceptionally affect the online learning atmosphere. Whenever the quality is high, students are in favour of using it, leading to positive evaluation by both students and teachers. This is due to the fact that the information is evaluated as sufficient and consistent. The information quality has a relation with the measurement of the excellence of the communication knowledge in content sources assessment [66–68]. Hence, it appears that both the education quality system and information quality can affect the future of online platforms. Therefore, it is hypothesised that:

**Hypothesis 6 (H6).** *The intention to use an online platform is positively affected by the education quality system.*

**Hypothesis 7 (H7).** *The intention to use an online platform is positively affected by the information quality.*

## 4. Research Methodology

*4.1. Data Collection*

Students studying in UAE universities were sent online surveys to gather data. The period chosen was the fall semester 2020/2021, from 17 February 2021 to 28 April 2021.

One thousand bottom of form (1000) questionnaires were randomly distributed by the researchers. Seven hundred and sixty-eight (768) questionnaires were responded to, indicating a response rate of 77%. Due to missing values, 232 filled questionnaires had to be rejected. As stated by [69], with valid responses for 768, the sample size was maintained as appropriate. This means that for a 1500 population, the respondent sample size was supposed to be 306. Yet, the sample size used, keeping in mind the minor requirements, was 768, which is much bigger. For the sample size, the analysis using the structural equation modelling is applicable [70], and this is much needed for hypothesis confirmation. Moreover, present theories have been used to establish the hypotheses; yet, they have been included within the e-learning context. The measurement model has been evaluated by the researchers through the application of SEM (SmartPLS Version 3.2.7, University of South Alabama, Mobile, AL, USA). Advanced treatment has been carried out using the final path model.

*4.2. Study Instrument*

In the current research, the hypothesis has been validated using a survey instrument. Eleven constructs have been measured in the questionnaire. There are 24 items that are

included in the survey. In Table 2, one can observe the constructs and their source. The applicability of this research study has been enhanced by altering and adjusting earlier research questions.

**Table 2.** Measurement Items.

| Constructs | Items | Instrument | Sources |
|---|---|---|---|
| Behavioural intention to use online platforms after the pandemic | BI1 | I am keen on continuously checking the online learning platform. | [71] |
| | BI2 | Overall, I am ready to use an online platform in the future. | |
| Educational system quality | ESQ1 | My online learning platform is collaborative and active. Therefore, I will use it even after the pandemic. | [72–74] |
| | ESQ2 | My online learning platform has a variety of learning styles. Therefore, I will use it even after the pandemic. | |
| | ESQ3 | My online learning platform has an interactive feature. Therefore, I will use it even after the pandemic. | |
| Information quality | IQ1 | My online learning platform provides me with up-to-date information. Therefore, I will use it even after the pandemic. | [72–74] |
| | IQ2 | My online learning platform provides me with the content I need at the right time. Therefore, I will use it even after the pandemic. | |
| | IQ3 | My online learning platform provides me with information that is easy to understand. Therefore, I will use it even after the pandemic. | |
| | IQ4 | My online learning platform provides me with organised content/information. Therefore, I will use it after the pandemic. | |
| Information richness | IR1 | My full understanding of the online platform urges me to keep using it after the pandemic. | [75] |
| | IR2 | Using an online platform after the pandemic will enhance my awareness of learning objectives and outcomes. | |
| | IR3 | My perception of new material is better if I continue using online platforms alongside face-to-face classes after the pandemic. | |
| Perceived ease of use | PEOU1 | I will continue using online platforms after the pandemic because it is easy to use them. | [76] |
| | PEOU2 | In my opinion, using an e-learning platform after the pandemic will be free of effort. | |
| | PEOU3 | Overall, using an online learning platform will be easy even after the start of the face-to-face classes. | |
| Users' situation awareness | USA1 | My clear vision of the material offered via online platforms helps me develop my learning skills. | [57,77,78] |
| | USA2 | Using an online platform after the pandemic will assist my persuasion and argumentation skills. | |
| | USA3 | My comprehension of new courses will be easier if online learning is still effective after the pandemic. | |

**Table 2.** *Cont.*

| Constructs | Items | Instrument | Sources |
|---|---|---|---|
| Perceived usefulness | PU1 | I will continue using online platforms after the pandemic because they are useful. | [76] |
| | PU2 | I will continue using online platforms after the pandemic because they help me complete different assignments and homework. | |
| | PU3 | I will continue using online platforms after the pandemic because they help in understanding my daily classes. | |
| Users' satisfaction | US1 | I will continue using online platforms after the pandemic because they satisfy my needs. | [79] |
| | US2 | I will continue using online platforms after the pandemic because they resolve my queries when I miss important information in face-to-face classes. | |
| | US3 | I will continue using online platforms after the pandemic because it fits my plans. | |

### 4.3. Pilot Study of the Questionnaire

A pilot study was carried out to check the reliability of the questionnaire item. As part of this pilot research, 100 students were randomly selected from the decided population. The entire sample size used in the research for the assessment should be 10%, and keeping this in mind, the sample size was 1000 students. The standard of research was maintained. The findings of the pilot study have been assessed by applying the Cronbach's Alpha (CA) test. It helps recognise the internal reliability through the IBM SPSS Statistics ver. 23 (IBM, Armonk, NY, USA). Hence, for all the measurement items, the conclusions presented were acceptable. The acceptable reliability coefficient is 0.70 when the stated social science research studies are considered [80]. Table 3 states the Cronbach alpha values for the five mentioned measurement scales.

**Table 3.** Cronbach's alpha values for the pilot study (CA $\geq$ 0.70).

| Constructs | CA |
|---|---|
| BI | 0.760 |
| ESQ | 0.785 |
| IQ | 0.878 |
| IR | 0.817 |
| PEOU | 0.881 |
| USA | 0.889 |
| PU | 0.825 |
| US | 0.816 |

Note: BI is behavioural intention to use online platforms after the pandemic; ESQ is educational system quality; IQ is information quality; IR is information richness; PEOU is perceived ease of use; USA is users' situation awareness; PU is perceived usefulness; US is users' satisfaction.

### 4.4. Survey Structure

The questionnaire survey was given to the students [81]. The survey contained three different sections.

- In the first section, the participant's data are recorded.
- In the second section, two items ask questions related to online learning platforms.

- In the third section, there are twenty-two items related to educational system quality, information quality, information richness, perceived ease of use, users' situation awareness, perceived usefulness and users' satisfaction. The five-point Likert scale has been used to measure the 24 items. The scale includes strongly disagree (1), disagree (2), neutral (3), agree (4) and strongly agree (5).

## 5. Findings and Discussion

### 5.1. Personal/Demographic Information

In Table 4, the personal/demographic information has been assessed and presented. The male-to-female ratio has been maintained at 40:60. Moreover, 33% of the respondents were above the age of 29 years, and 67% of the respondents were between 18 and 29 years in age. An educated background with a university degree was associated with most respondents. In the sample, 65% of the respondents attained a bachelors' degree, 23% had a masters' degree and 12% had a doctorate degree. When respondent access is easy and they volunteer willingly, then the purposive sampling approach should be implemented [82–84]. The research sample was developed using students from various colleges. The age of these students is different and their programs and levels all vary. Furthermore, the IBM SPSS Statistics ver. 23 was applied to measure the demographic data. Table 4 indicates thorough respondent demographic data.

**Table 4.** Demographic data of the respondents.

| Criteria | Factor | Frequency | Percentage |
|---|---|---|---|
| Gender | Female | 460 | 60% |
| | Male | 308 | 40% |
| Age | Between 18 and 29 | 516 | 67% |
| | Between 30 and 39 | 137 | 18% |
| | Between 40 and 49 | 83 | 11% |
| | Between 50 and 59 | 32 | 4% |
| Education qualification | Bachelors' | 498 | 65% |
| | Masters' | 177 | 23% |
| | Doctorate | 93 | 12% |

### 5.2. Data Analysis

By applying the SmartPLS V.3.2.7 software and the partial least squares–structural equation modelling (PLS-SEM), the research study data analysis was carried out [85]. The collected data were assessed through the application of an assessment approach that has two steps: a structural model and a measurement model [86]. Within the current research, the PLS-SEM has been applied for two reasons.

At first, the most appropriate choice is the PLS-SEM since the current research requires for the existing theory to be built [87]. Second, the PLS-SEM can be applied to efficiently manage the exploratory research attaining complex models [88]. Third, the PLS-SEM does not divide the entire model into fragments but assesses it as a whole [89]. Fourth, the PLS-SEM carries out a concurrent analysis for the measurement and structural model. Hence, the calculations attained are precise [90].

### 5.3. Convergent Validity

The authors [86] recommend that when the measurement model is assessed, the validity, which includes convergent and discriminant validity, and construct reliability, which includes composite reliability (CR), Dijkstra–Henseler's (PA) and CA, should be taken into account. According to Table 5, construct reliability can be determined using CA with values between 0.730 and 0.833. The threshold value is 0.7, and the mentioned

figures are higher [80]. Table 5 also indicates that the CR attains values between 0.770 and 0.904, and these are also higher than 0.7, which is the recommended value [91]. By applying the Dijkstra–Henseler's rho (pA) reliability coefficient, researchers must assess as well as report the construct reliability. Like CR and CA, values of 0.70 or higher should be indicated by the reliability coefficient pA as part of the exploratory research [66]. If the research is expected to be more advanced, then values should be over 0.80 or 0.90 [80,92,93]. It has also been observed in Table 5 that for each measurement construct, the reliability coefficient pA must be over 0.70. Based on the mentioned results, there is confirmation for construct reliability, and towards the end, the constructs are assumed to be free of error in a sufficient manner. Convergent validity should be measured by testing the average variance extracted (AVE) and factor loading [86]. The findings in Table 4 indicate that the suggested value of 0.7 was always lower than all factor-loading values. Furthermore, according to Table 5, values between 0.540 and 0.758 were produced by AVE, and these are higher than the 0.5 threshold value. Keeping in mind the future results, convergent validity can be attained successfully for all constructs.

**Table 5.** Convergent validity results that assure acceptable values (factor loading, CA, CR ≥ 0.70 and AVE > 0.5).

| Constructs | Items | Factor Loading | CA | CR | PA | AVE |
|---|---|---|---|---|---|---|
| Behavioural intention to use online platforms after the pandemic | BI1 | 0.822 | 0.829 | 0.897 | 0.829 | 0.745 |
| | BI1 | 0.729 | | | | |
| Educational system quality | ESQ1 | 0.754 | 0.765 | 0.844 | 0.779 | 0.679 |
| | ESQ2 | 0.733 | | | | |
| | ESQ3 | 0.855 | | | | |
| Information quality | IQ1 | 0.848 | 0.778 | 0.870 | 0.788 | 0.692 |
| | IQ2 | 0.777 | | | | |
| | IQ3 | 0.910 | | | | |
| Information richness | IR1 | 0.859 | 0.777 | 0.770 | 0.653 | 0.540 |
| | IR2 | 0.904 | | | | |
| | IR3 | 0.891 | | | | |
| Perceived ease of use | PEOU1 | 0.874 | 0.803 | 0.884 | 0.802 | 0.717 |
| | PEOU2 | 0.853 | | | | |
| | PEOU3 | 0.822 | | | | |
| Users' situation awareness | USA1 | 0.771 | 0.730 | 0.850 | 0.738 | 0.654 |
| | USA2 | 0.828 | | | | |
| | USA3 | 0.890 | | | | |
| Perceived usefulness | PU1 | 0.781 | 0.761 | 0.866 | 0.770 | 0.609 |
| | PU2 | 0.858 | | | | |
| | PU3 | 0.864 | | | | |
| Users' satisfaction | US1 | 0.880 | 0.833 | 0.904 | 0.846 | 0.758 |
| | US2 | 0.836 | | | | |
| | US3 | 0.871 | | | | |

## 5.4. Discriminant Validity

Measurement of two criteria, Fornell–Larcker and the Heterotrait–Monotrait ratio (HTMT), has been recommended for the discriminant validity measurement [86]. The outcomes of Table 6 indicate that the requirements of the Fornell–Larcker condition are

confirmed since the AVEs and their square roots are higher than the rest of the correlation constructs [94].

**Table 6.** Fornell–Larcker Scale.

|       | BI    | ESQ   | IQ    | IR    | PEOU  | USA   | PU    | US    |
|-------|-------|-------|-------|-------|-------|-------|-------|-------|
| BI    | **0.798** |       |       |       |       |       |       |       |
| ESQ   | 0.450 | **0.872** |       |       |       |       |       |       |
| IQ    | 0.692 | 0.363 | **0.885** |       |       |       |       |       |
| IR    | 0.626 | 0.538 | 0.296 | **0.880** |       |       |       |       |
| PEOU  | 0.505 | 0.065 | 0.237 | 0.601 | **0.856** |       |       |       |
| USA   | 0.444 | 0.500 | 0.573 | 0.592 | 0.451 | **0.817** |       |       |
| PU    | 0.458 | 0.583 | 0.553 | 0.476 | 0.513 | 0.307 | **0.851** |       |
| US    | 0.446 | 0.565 | 0.641 | 0.616 | 0.604 | 0.391 | 0.521 | **0.844** |

Note: BI is behavioural intention to use online platforms after the pandemic; ESQ is educational system quality; IQ is information quality; IR is information richness; PEOU is perceived ease of use; USA is users' situation awareness; PU is perceived usefulness; US is users' satisfaction.

Table 7 indicates the results for the HTMT ratio and shows that, for each construct value, the 0.85 threshold value stays ahead [95]. Therefore, the HTMT ratio is created. Based on the mentioned findings, the discriminant validity is stated. Keeping in mind the results of the analysis, the measurement model assessment did not have any concerns in terms of reliability and validity. Hence, it is possible to assess the structural model by applying the collected data.

**Table 7.** Heterotrait–Monotrait Ratio (HTMT).

|       | BI    | ESQ   | IQ    | IR    | PEOU  | USA   | PU    | US    |
|-------|-------|-------|-------|-------|-------|-------|-------|-------|
| BI    |       |       |       |       |       |       |       |       |
| ESQ   | 0.232 |       |       |       |       |       |       |       |
| IQ    | 0.202 | 0.517 |       |       |       |       |       |       |
| IR    | 0.260 | 0.681 | 0.611 |       |       |       |       |       |
| PEOU  | 0.506 | 0.633 | 0.609 | 0.333 |       |       |       |       |
| USA   | 0.243 | 0.392 | 0.111 | 0.144 | 0.255 |       |       |       |
| PU    | 0.501 | 0.658 | 0.753 | 0.511 | 0.721 | 0.512 |       |       |
| US    | 0.207 | 0.672 | 0.511 | 0.419 | 0.290 | 0.463 | 0.721 |       |

Note: BI is behavioural intention to use online platforms after the pandemic; ESQ is educational system quality; IQ is information quality; IR is information richness; PEOU is perceived ease of use; USA is users' situation awareness; PU is perceived usefulness; US is users' satisfaction.

*5.5. Model Fit*

The fit measures offered by SmartPLS are standard root mean square residual (SRMR), exact fit criteria, d_ULS, d_G, Chi-Square, NFI and RMS_theta, which show the PLS-SEM model fit [96]. The difference present amongst the observed correlations and model implied correlation matrix [88] is denoted by SRMR, and the good model fit measures are values lower than 0.08 [97]. A good model fit is the NFI values that are over 0.90 [98]. The ratio of the proposed model Chi2 value to the benchmark or null model is the NFI [99]. The NFI and parameters have a positive association, which is why the NFI is not considered to be a model fit indicator [88]. The empirical covariance matrix and covariance matrix discrepancy can be observed in two metrics, which are squared Euclidean distance, d_ULS, and the geodesic distance d_G. This has been implied using the composite factor model [88,100]. For the reflective model, only the RMS theta is applied, and the correlation degree of the

outer model residuals is evaluated [99]. The PLS-SEM model is more efficient when the RMS theta value is closer to zero, and it would be a good fit if the value is lower than 0.12. There would be a lack of fit for anything else [101]. With the help of the saturated model, the correlation amongst the constructs is assessed, and the model structure and total effects are observed by the estimated model [88].

According to Table 8, 0.069 is the RMS_theta value, and it shows that the goodness-of-fit for the PLS-SEM model is appropriate enough to indicate the validity of the global PLS model.

**Table 8.** Model fit indicators.

| | Complete Model | |
| --- | --- | --- |
| | **Saturated Model** | **Estimated Model** |
| SRMR | 0.066 | 0.066 |
| d_ULS | 0.770 | 1.538 |
| d_G | 0.503 | 0.503 |
| Chi-Square | 477.558 | 477.558 |
| NFI | 0.685 | 0.685 |
| Rms Theta | 0.069 | |

*5.6. Hypotheses Testing Using PLS-SEM*

Through Smart PLS, it was possible to use the SEM, and maximum likelihood estimation was present to enable the recognition of the interdependence of the structural model and several theoretical constructs [102–106]. Similarly, it was possible to assess the proposed hypotheses. Figure 2 and Table 9 indicate that there is moderate predictive power in the model [107], which means that 55.7% is the variance percentage for the behavioural intention to use online platforms after the pandemic.

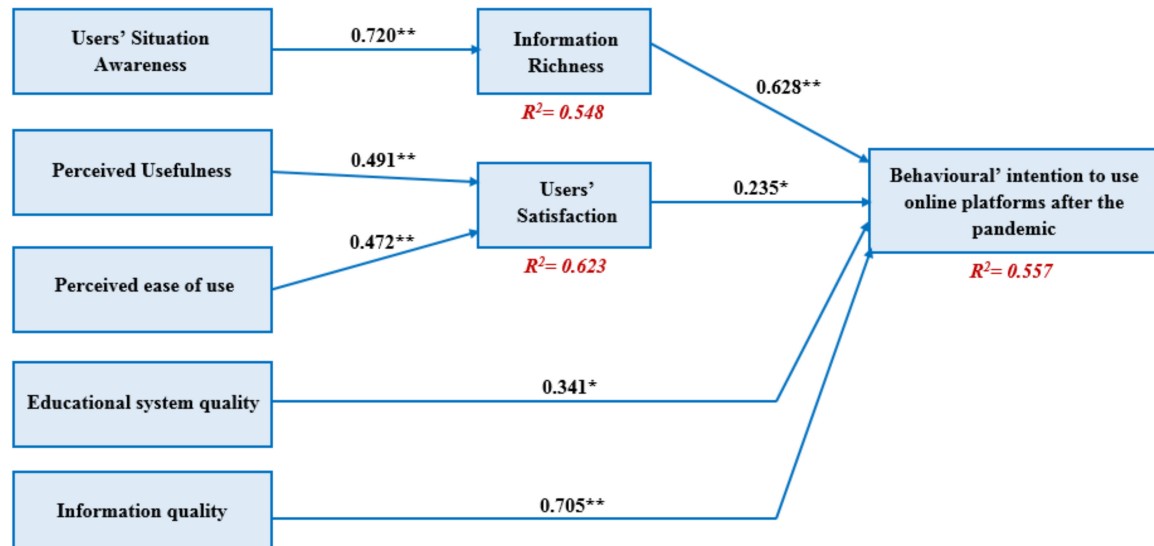

**Figure 2.** Path coefficient of the model (significant at ** $p \leq 0.01$, * $p < 0.05$).

**Table 9.** $R^2$ of the endogenous latent variables.

| Constructs | $R^2$ | Results |
|:---:|:---:|:---:|
| BI | 0.557 | Moderate |
| IR | 0.548 | Moderate |
| US | 0.623 | Moderate |

Note: BI is behavioural intention to use online platforms after the pandemic; information richness; users' satisfaction.

For each stated hypothesis, the beta ($\beta$) values, *t*-values and *p*-values are stated in Table 10, and these are extracted using the PLS-SEM technique. It has been observed that all the hypotheses have been supported by the researchers. Considering the data assessment, empirical data support the hypotheses H1, H2, H3, H4, H5, H6 and H7. The relationships between users' situation awareness (USA) and information richness (IR) ($\beta = 0.720$, $p < 0.001$) were found to be statistically significant, and thus, hypothesis H1 is generally supported. The results showed that users' satisfaction (US) significantly influences perceived ease of use (PEOU) ($\beta = 0.491$, $p < 0.001$) and perceived usefulness (PU) ($\beta = 0.472$, $p < 0.001$), supporting hypotheses H3 and H4, respectively. Furthermore, information richness (IR), users' satisfaction (US), educational system quality (ESQ) and information quality (IQ) have significant effects on behavioural intention to use online platforms after the pandemic (BI) ($\beta = 0.628$, $p < 0.001$), ($\beta = 0.235$, $p < 0.05$), ($\beta = 0.341$, $p < 0.05$) and ($\beta = 0.705$, $p < 0.001$), respectively; hence, H2, H5, H6 and H7 are supported.

**Table 10.** Hypotheses testing of the research model (significant at ** $p \leq 0.01$, * $p < 0.05$).

| H | Relationship | Path | *t*-Value | *p*-Value | Direction | Decision |
|:---:|:---:|:---:|:---:|:---:|:---:|:---:|
| H1 | USA -> IR | 0.720 | 28.657 | 0.000 | Positive | Supported ** |
| H2 | IR -> BI | 0.628 | 10.880 | 0.000 | Positive | Supported ** |
| H3 | PU -> US | 0.491 | 15.489 | 0.000 | Positive | Supported ** |
| H4 | PEOU -> US | 0.472 | 15.228 | 0.003 | Positive | Supported ** |
| H5 | US -> BI | 0.235 | 3.277 | 0.029 | Positive | Supported * |
| H6 | ESQ -> BI | 0.341 | 3.454 | 0.031 | Positive | Supported * |
| H7 | IQ -> BI | 0.705 | 8.072 | 0.000 | Positive | Supported ** |

Note: BI is behavioural intention to use online platforms after the pandemic; ESQ is educational system quality; IQ is information quality; IR is information richness; PEOU is perceived ease of use; USA is users' situation awareness; PU is perceived usefulness; US is users' satisfaction.

## 6. Discussion of the Results

The data analysis has shown that all the seven hypotheses are supported, which empowers our assumptions. The information richness, educational system quality, information quality, satisfaction and TAM constructs have a decisive role in measuring students' perception of using online learning platforms. The crucial issues that can be highlighted are related to the association between satisfaction and PEOU and PU on one hand and that between information richness and SA on the other hand, as is explained below.

The current results are in line with prior studies regarding the crucial role of information richness in measuring the intention to use online learning platforms, and they support the awareness situation with the three dimensions of perception, comprehension and projection. These three dimensions play a major role in the acceptance of online platforms [108]. The content richness can enrich the online platform pedagogically by adding the features of interactivity, convenience, instant feedback and social learning. This implies that the higher the content richness is, the more effective the online platform will be [108–110].

The secondary result is concerned with the importance of the correlation between users' satisfaction and the two TAM constructs. The higher the degree of perceived ease of use and the perceived usefulness is, the higher the satisfaction level will be. The prior conclusion is consistent with the argument that is proposed by studies from [111–113]. These studies focused on the effectiveness of TAM in different fields such as the agricultural and oil industries and arrived at the conclusion that TAM factors have a varying degree of effectiveness and can significantly affect the acceptance of online learning platforms virtually and practically. The results of the study show that the research model has been validated successfully. The perceived ease of use has a significant effect on the intention of using an online learning platform from students' perspectives. The findings also reveal that perceived usefulness has a significant effect on students' intentions to use this technology and can lead to successive actual use. It explains why students are in favour of constantly using online learning platforms, as they are advantageous in terms of enhancing and facilitating the teaching–learning process.

The third result is concerned with the educational system quality and information quality, which are crucial factors that affect the intention to use online platforms. A student's perception of an online learning platform will be more satisfactory whenever they have a high education quality system and effective information quality [114,115]. Accordingly, system quality will significantly affect the intention to use the online learning platform. The chances to use the online platform will increase if the system is annually improved. This leads to a positive evaluation by students, and it will tremendously affect their perception of the system. In short, when the system quality is improved continuously, it will significantly and positively affect students' perception, especially when the improvement is related to basic features that keep immediate feedback, online communication and accessibility active. The information quality will be affected by such types of improvement, leading to a positive and significant impact on students' perception. The current result is in agreement with previous studies that stated that information quality is considered significant if it satisfies users' needs [116–118]. Both the system quality and information quality of online learning platforms have a significant effect on the intention to use an online learning platform. This result is consistent with [59,117,119,120] and others who stated that the effectiveness of system quality and information quality are significant and the improvement of both will result in more fruitful efforts.

In conclusion, this study has proven that the effectiveness of e-learning platforms will be evident after the pandemic. The fact that e-learning platforms offer ease of use and usefulness will help students and teachers to continue using them even when face-to-face classes are resumed. These two features increase the degree of satisfaction expressed by users. The other variables of SA and information flow have added more advantages to using these platforms. Pedagogical and academic factors are highly affected by the use of these platforms in the educational environment. Finally, when the system quality and the information quality adequately meet the users' needs, the users' perception is improved. Thus, the study has shown that e-learning platforms are influential means of teaching along with traditional classes due to their specific and unique features.

### 6.1. Theoretical and Practical Implications

The results obtained from the current study can serve teachers, students and educational system developers both theoretically and practically. Theoretically speaking, the study provides short and reliable recommendations that may help measure the acceptance level of technology amongst a group of students. Practically speaking, the findings will assist online learning platform developers and designers to provide users with a friendly and fruitful interface where every possible feature can contribute to the teaching–learning environment The effectiveness stems from the fact that they can offer tools and facilities that drive them to seek information by depending on their online learning platform more regularly.

### 6.2. Managerial Implications

The study confirms that heads of universities and colleges are capable of developing their online learning platform in a way that encourages their users to make use of the system consistently. Based on their proposed assumptions, it is observed that system improvement is fruitful, and this may be beneficial from a marketing and revenue perspective. The findings of the study highlight that the intention to use online learning platforms should be carefully monitored by system managers and the IT-support system.

### 6.3. Limitations and Suggestions for Future Studies

Regardless of our contributions to the literature by creating a comparison of specific models and external factors used in prior studies, outlining the type of technology and technological model, the research study has some limitations. First, the selected studies are limited to a particular group that spans periods of time, one before the pandemic and one after the outbreak of the pandemic. Future studies can extend the period to include studies in the future when the effect of the pandemic declines.

Second, the conceptual model is restricted to factors that affect students' acceptance with respect to information richness, TAM constructs, system quality and information quality. Future studies can include other factors such as attitude, perceived enjoyment, perceived security and perceived addiction.

Third, the samples are limited to a group of students in the Gulf area. Future studies can compare the attitudes of students from the Gulf area and other places such as Malaysia, China, the UK, etc. Hence, the results may vary from the current ones.

Fourth, the obtained results are based on a questionnaire that has close-ended types of questions. Future works should consider using a questionnaire that is qualitative and employ a data collection method that is based on interviews or observations.

**Author Contributions:** Conceptualization, I.A.A.; formal analysis, M.A. and S.S.; investigation, S.T.; methodology, K.A. (Khadija Alhumaid) and R.A.; project administration, S.S.; resources, N.M.A.; software, N.A.; supervision, K.A. (Kevin Ayoubi); visualization, A.A.; writing—original draft, R.S.A.-M.; writing—review and editing, S.S. All authors have read and agreed to the published version of the manuscript.

**Funding:** This research received no external funding.

**Institutional Review Board Statement:** Not applicable.

**Informed Consent Statement:** Not applicable.

**Data Availability Statement:** The data presented in this study are available on request from the corresponding author.

**Conflicts of Interest:** The authors declare no conflict of interest.

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
