# Peer review of "The Effectiveness of Online Platforms after the Pandemic: Will Face-to-Face Classes Affect Students’ Perception of Their Behavioural Intention (BIU) to Use Online Platforms?"

_informatics, doi:10.3390/informatics8040083_

Round 1
Reviewer 1 Report
This is a well-structured paper. The research topic is interesting and the research question is meaningful. Yet the perspective that the authors used to examine the phenomenon needs further explanation and clarification. As shown in Table 1, the TAM and UTAUT have been repeatedly used by a number of studies. Why is it necessary to test it again and what would be the implications by doing so? The rationale behind linking TAM with the theory of situation awareness needs to be strengthened in the introduction section. Why is the theory of situation awareness relevant in this context?
The dimension “educational system quality” seems to be overlapping with intention to use. Why do all ESQ dimensions end with “therefore I will use it even after the pandemic”? Same for IQ, IR, PEOU, PU, and US measurements, they all ended with “therefore I will use it even after the pandemic”. These are double-barreled questions.
The theoretical foundation of the theory of situation awareness is unclear. How were the variables “educational system quality”, “information quality”, “perception comprehension projection”, and “information richness” developed based upon the theory of situation awareness? Were these variables all come from the same origin/source?
Why included all three dimensions in one single hypothesis (H1). The expressions of some of the hypotheses are incorrect (H1, H3, H4). Please correct them so they are consistent with the research model (Figure 1).
I hesitate to recommend this paper for publication due to the unclear theoretical foundation of the conceptual model, the problematic questionnaire design, and the limited theoretical contributions.
Author Response
The authors are really very grateful to the feedback and comments raised by the reviewer which really assist them to significantly enhance this work and its presentation. The productive and valuable remarks enable us to update many parts of the paper as shown by the responses to each comment. Our responses are mentioned below under each comment raised by the reviewer and it is written in (Times New Roman, red color). Besides, all the updated parts in the manuscript were highlighted in yellow color in order to be easily tracked by the reviewers.

Reviewer 2 Report
Please see attached.

Author Response

(The authors gave the same response as above.)

Reviewer 3 Report
It would be nice to include in the Introduction the specific objectives in relation to the findings they have mentioned; How did your results address the problem?
Provide more details on defining variables. It would be nice to include the mathematical formulas and references for the different calculations.
It is better to provide a more up-to-date bibliographic review. There are excellent, very current examples published on the subject of Information Journal. The article has no reference from the journal Information. Journal Information has excellent references related to the topic. Even different journal of MDPI have published excellent works related to the subject.
For example:
https://www.mdpi.com/2071-1050/13/7/3820
https://www.mdpi.com/2071-1050/12/18/7514
Author Response

(The authors gave the same response as above.)

Reviewer 4 Report
The paper addresses a topic of interest by testing an explanatory model of the intention to use e-learning platforms after an exclusive period of use in the learning process.
The manuscript is well structured, the methodology is appropriate to test the assumptions, and the conclusions are supported by results.
I have just one suggestion. On line 78 it is written: "This study is motivated by the need to fill this knowledge gap". I think it is necessary to write something in the conclusions section about this gap. By testing this model, have you discovered something new compared to what was known before? If so, what is different? If not, it would be useful to mention this and explain these results.
Author Response

(The authors gave the same response as above.)

Round 2
Reviewer 2 Report
I thanked for the author's response. However, most of the comment that has previously been made was not properly addressed and clarified.
Author Response

(The authors gave the same response as above.)
